# Subjective Well-Being and Related Factors among Independent, Healthy, Community-Dwelling Older Adults in Japan

**DOI:** 10.3390/healthcare11152211

**Published:** 2023-08-06

**Authors:** Yuki Imamatsu, Nanami Oe, Eriko Ito, Yuka Iwata, Azusa Arimoto, Kisaki Kobayashi, Etsuko Tadaka

**Affiliations:** 1Department of Community Health Nursing, Graduate School of Medicine, Yokohama City University, Fukuura 3-9, Kanazawa-Ku, Yokohama 236-0004, Japan; imamatsu.yuk.db@yokohama-cu.ac.jp (Y.I.);; 2Department of Community and Public Health Nursing, Graduate School of Health Sciences, Hokkaido University, K12-N5, Kita-Ku, Sapporo 060-0812, Japan; o_nanami0706@pop.med.hokudai.ac.jp (N.O.);

**Keywords:** WHO-5, population, social network, psychological well-being, older adult

## Abstract

(1) Background: Subjective well-being (SWB) is internationally recognized as an important health-related factor for most age groups and is particularly influential for life quality and expectancy in independent, healthy, community-dwelling older adults. However, the physical function and community participation correlates of SWB in independent living older adults in super-aging societies and other influencing factors remain underexplored. (2) Methods: A total of 926 independent, healthy, community-dwelling older adults aged 65 years and above registered in Yokohama, Japan, were included. Respondents’ mean age was 78.1 years (standard deviation = 6.7), and 74.0% were women. The dependent variable was SWB. The independent variables were respondents’ demographic characteristics, physical factors (visual, hearing, and cognitive functions, and mobility), and community and social factors (participation in community groups, social networks, and community commitment. (3) Results: The mean (standard deviation) WHO-5 score was 16.3 (5.1). Significant factors associated with WHO-5 score were visual function (odds ratio [OR]: 0.708; 95% confidence interval [CI]: 0.352–0.690), hearing function (OR: 0.615; CI: 0.431–0.878), community groups (OR: 1.310; CI: 1.003–1.059), community commitments (OR: 1.180; CI: 1.132–1.231), and social networks (OR: 1.525; CI: 1.142–2.037) adjusted for the effects of demographic factors. (4) Conclusions: These findings are important because factors associated with SWB are likely to contribute to individual well-being and longevity and to developing a healthy super-aged society.

## 1. Introduction

Japan has the highest rate of population aging among super-aged nations worldwide [1], with a current average life expectancy of 81.9 years for men and 87.3 years for women. Half of all older Japanese men are above the age of 84, with more than 25% of men in this group reaching an age above 90 years. Half of older Japanese women are above the age of 90, with more than 25% of women in this group living longer than 95 years. Moreover, the average life expectancies for both men and women are projected to continue to rise, reaching approximately 85.0 years for men and approximately 91.0 years for women by 2065 [2]. Furthermore, it is projected that in the future, a life expectancy of 100 years will become the norm in Japan. The Japanese government is developing a “100-year life expectancy strategy”, which includes policy design for creating a social system that accommodates a 100-year lifespan. Given the global trend of population aging, an urgent challenge facing the international community entails fostering a society in which older people can lead healthy and active lives with a sense of well-being regardless of their health status. In particular, preventive strategies to support the subjective well-being (SWB) [3] of independent, healthy, older adults is a public health priority for Japan and other developed countries with aging populations.

SWB is a self-reported measure of well-being. The first SWB model, which was developed in 1984 by Diener [3], proposed a framework for understanding how individuals experience the quality of their lives. SWB and health are closely related [4], and this link may become increasingly important at older ages, if only because the prevalence of chronic illness increases with advancing age. Previous studies have reported evidence that higher SWB has a protective or positive effect on the maintenance of good health among older adults [5,6,7,8]. An analysis of the English Longitudinal Study of Ageing reported that SWB is associated with increased survival; 29.3% of people in the lowest well-being quartile died during the average follow-up period of 8.5 years, compared with 9.0 3.0% of those in the highest quartile. Associations were independent of age, sex, demographic factors, and baseline mental and physical health [5]. Moreover, another meta-analysis showed that SWB was a protective factor for mortality (pooled hazard ratio = 0.920; 95% confidence interval [CI] = 0.905–0.934) [9]. Thus, SWB is one of the most important health outcomes for older adults.

A systematic review [10] reported that older adults with a high level of SWB may be less negatively affected by ageism, particularly for those who are proud of their age group, who experience fewer negative emotions, are more optimistic about aging and their future, are more self-confident about their bodies, and are flexible in setting goals. A systematic review on the determinants of SWB in older adults have identified physical function and physical activity, mental activity, life satisfaction, leisure satisfaction [11], social relationships, social network, community activities, and social capital as influential factors for SWB [12,13]. Although various factors of SWB in older adults have already been presented, any factors of SWB cannot be determined unless confounding variables are taken into account, including adjustment for demographic factors. Moreover, better understanding of the influence of these factors could provide a basis for enhanced health care and may benefit older adults. Therefore, a comprehensive understanding perspective is crucial for SWB and related factors among older adults.

The International Classification of Functioning, Disability and Health (ICF) is a classification of health and health-related domains [14]. The ICF was officially endorsed by all 191 World Health Organization (WHO) Member States in the 54th World Health Assembly in 2001 as the international standard for describing and measuring health and disability. The domains contained in the ICF can be considered as health domains and health-related domains. These domains are described from the perspective of the body, the individual, and society, in two basic lists: (1) body functions and structures; and (2) activities and participation. In principle, the ICF structures the comprehensive universe of well-being, which allows researchers to generate hypotheses about the relationships among concepts and variables. On the basis of this theory, therefore, it can be hypothesized that SWB is related to individual physical factors that may directly or indirectly interact with body functions and structures and community and social factors that may directly or indirectly interact with activities and participation. However, the associations between SWB and individual physical factors and community and social factors among older adults living in the community remain underexplored.

Aging is influenced by physical, social, and mental well-being, participation in society, protection, security, and care. If older adults are able to be physically active and continue participation in social, economic, cultural, spiritual, and civil affairs, it is active healthy aging [14]. Moreover, active healthy ageing is the process of optimizing opportunities for successful aging [15]. That is, SWB and its related individual physical factors as well as community and social factors can be a powerful tool for achieving successful aging [15]. Therefore, the purpose of this study was to clarify the associations between SWB and individual physical factors, as well as community and social factors, among independent older adults living in the community with an average age of 75 or older, which is expected to increase rapidly in the future. We tested the following hypotheses.

**Hypothesis** **1.**
*The individual physical factors, such as visual dysfunction, hearing dysfunction, and cognitive dysfunction, which are included as physical functions and structures in the ICF, are more negatively associated with SWB among independent older adults living in the community, even after adjusting for demographic factors.*


**Hypothesis** **2.**
*The community and social factors, such as community group participation, social networks, and community commitment, which are included as activities and participation in ICF, are more positively associated with SWB among independent older adults living in the community, even after adjusting for demographic factors.*


## 2. Materials and Methods 

### 2.1. Study Participants and Design

The inclusion criteria for this study participants were as follows: (1) individuals aged 65 years and above; (2) older adults certified by the local government as being independent, healthy, and living in communities; and (3) individuals with the ability and willingness to complete the self-administered questionnaire. The exclusion criteria were individuals under the age of 65 years or of unknown age. The survey was given to a purposive sample of 1900 independent, healthy, community-dwelling older adults aged 65 years and above registered in Yokohama, the largest city designated by government ordinance in Japan, of whom 1154 responded (a response rate 60.8%). Of these individuals, 228 people were excluded according to our exclusion criteria. Thus, the final sample for the analysis comprised 926 older adults. Table 1 shows the demographic characteristics of the 926 participants. The mean age of the respondents, 74% of whom were women, was 78.1 years (standard deviation [SD] = 6.7, range 65–100 years). Of the participants, 28.5% lived alone and 35.9% lived with a spouse. Moreover, 10.0% were employed, and 86.4% perceived their economic status to be sufficient.

### 2.2. Measures

#### 2.2.1. Dependent Variables

The dependent variable in this study was SWB. The Japanese version of the Five-Item World Health Organization Well-Being Index (WHO-5) was used as a measure of SWB [16]. The WHO-5 is one of the most widely used questionnaires for assessing SWB internationally. It has been translated into various languages (see http://www.who-5.org/, accessed on 4 August 2023), and its use in the context of various health conditions [17] and to assess health-related quality of life (QOL) [16,17], as well as SWB, has been validated. A particular advantage of this scale is that it has sufficient internal and external validity for detecting mental health issues within the general population of older adults. Each item in the scale measures positive well-being [18] and is readily accepted by older community-dwelling adults, regardless of educational background or age. This scale comprises five items that measured respondents’ perceptions in the last 2 weeks: (1) “I have felt cheerful and in good spirits”. (2) “I have felt calm and relaxed”. (3) “I have felt active and vigorous”. (4) “I woke up feeling fresh and rested”. (5) “My daily life has been filled with things that interest me”. Each of these items were scored using a 6-point Likert scale. Scores ranged from 0 to 25, with higher scores indicating better levels of current mental health. The values of Cronbach’s α and Loevinger’s coefficient were 0.87 and 0.64, respectively. The total scores were significantly correlated with the number of cohabitants, the number of physical illnesses affecting the respondent, their physical functioning, instrumental activities of daily living, and depressive symptoms. A standard cut-off criterion of “a total score ≤ 12 or answering 0 or 1 to any of the five items” is considered to be more appropriate for identifying older adults with suicidal ideation: sensitivity = 87% and specificity = 75% [19].

#### 2.2.2. Demographic Characteristics

The demographic characteristics considered in the study were age, sex, living arrangements (living alone/with a spouse/with children/with a spouse and children/with children and grandchildren/other), economic status (0 = subjectively sufficient economic status, and 1 = subjectively insufficient economic status), and medical treatment history (International Statistical Classification of Diseases and Related Health Problems-10th revision: yes/no) [20].

#### 2.2.3. Physical Factors

Physical factors included visual function (self-reported visual impairment), hearing function (self-reported hearing impairment), cognitive function (measured with the Self-administered Dementia Checklist [SDC] [21]), and mobility (measured as the number of days per week when the respondent went outside). Cognitive decline was measured using the SDC. This scale comprises 10 items, each of which is scored from 10 to 40 using a 4-point Likert scale. Higher scores indicate lower cognitive function. The value of Cronbach’s α was 0.908 [22]. The SDC scores were significantly correlated with respondents’ Mini-Mental State Examination (MMSE) (r = −0.536, *p* < 0.001) [23] and Frontal Assessment Battery scores (FAB) (r = −0.457, *p* < 0.001) [24]. The MMSE and the FAB are both widely used screening tests for cognitive impairment in older adults. MMSE scores range from 0–30 and FAB scores range from 0–18, with higher scores on both scales indicating higher cognitive function. In this study, the cutoff point for cognitive impairment was 17/18 SDC items (sensitivity = 72.0% and specificity = 69.2) and was used to determine the presence/absence of mild cognitive impairment [21].

#### 2.2.4. Community and Social Factors

Community and social factors considered in this study were community group participation (1 = presence/0 = absence), social networks, and community commitment. Respondents’ social networks were measured using the Lubben Social Network Scale (LSNS-6) [25,26]. This scale comprises six items that measure the size, closeness, and frequency of contact with family members and friends within each respondent’s social network. All of the items are measured using a six-point Likert scale, with higher scores (within a range of 0 to 30) indicating more extensive social networks. The value of Cronbach’s α was 0.82 [25]. The mean LSNS-6 score increased with the number of households living together (*p* = 0.033), the suicide risk group had lower mean scores than the no suicide risk group (*p* = 0.026), and the poor subjective health status group tended to have lower mean scores than the good subjective health group (*p* = 0.081). Community commitment was measured using the Community Commitment Scale (CCS) [27]. This scale comprises eight items, divided into subscales of socializing and belonging, each of which has four items. Each item is scored using a four-point Likert scale, with higher scores (within a range of 0 to 24) indicating a stronger degree of community commitment. Cronbach’s α values for the CCS were 0.75 for local volunteers and 0.78 for general residents. The correlation coefficients between the scores of the CCS and the Brief Sense of Community Scale, with higher scores (within a range of 0 to 32) indicating a high level of sense of community [28,29], were 0.54 (local volunteers) and 0.62 (general residents).

### 2.3. Statistical Analysis

We applied frequencies and percentages for categorical variables and means (SD) for continuous variables within our descriptive statistical analyses. Determinants of WHO-5 scores were assessed through logistic regression analyses. In the logistic regression analysis of physical factors, we adjusted the covariates for age effects (1: less than mean age = 78.1 0: more than mean age), and in the logistic regression analysis of community and social factors, we adjusted the covariates for the effects of sex and physical factors. A *p*-value < 0.05 or a 95% CI that did not include 1 indicated that the results were statistically significant. The SPSS software package (version 28.0; IBM Corp., Armonk, NY, USA) was used for the analysis.

### 2.4. Ethical Considerations

Participants were informed, both in writing and verbally, of the study’s purpose and methods, and were informed that refusal to participate in the study or subsequent withdrawal would not have any negative implications for them, in the Yokohama City community health center, through city officials. We also explained that participation was voluntary and that completing and returning the questionnaire indicated their consent to participate in the study. This research was conducted in accordance with the 1964 Declaration of Helsinki (and its amendments), and the ethical guidelines for life sciences and medical research involving human subjects presented by the Ministry of Health, Labour and Welfare of Japan. The study was approved by the institutional ethical review board of the School of Medicine, Yokohama City University (approval no. A201200008).

## 3. Results

The mean (SD) WHO-5 scale score of the 926 participants was 16.3 (5.1). Table 2 shows physical and community and social factors of participants. Of the physical factors, subjective visual impairment and subjective hearing impairment were, respectively, reported by 20.4% and 17% of participants. The mean (SD) SDC scores were 17.3 (6.2), and the scores of 30.8% of participants were below the threshold indicating cognitive impairment. Of the community and social factors, community group participation was reported by 31.6% of respondents, and the mean (SD) values of the LSNS-6 and CCS scores were 17.2 (3.7) and 17.5 (6.0), respectively. 

A univariate Spearman’s correlation analysis of WHO-5 scores with demographic, physical, and community and social factors showed a significant negative correlation with age (*p* < 0.01), with WHO-5 scores decreasing with advancing age. Among the physical factors, there were significant negative correlations with disabilities associated with visual function (*p* < 0.001), hearing function (*p* < 0.001), and cognitive function (*p* < 0.01). For community and social factors, the presence of community groups (*p* < 0.01), scores for community commitment (*p* < 0.01), and scores for social networks (*p* < 0.001) exhibited significant positive correlations with WHO-5 scores.

In light of the above results and taking multicollinearity into consideration, we entered age and sex as control variables into the logistic regression analysis. Table 3 shows related factors in the logistic regression of the WHO-5 scores. Physical factors, adjusted for the effects of demographic factors, that were significantly associated with WHO-5 scores were visual function (odds ratio [OR]: 0.708; CI: 0.352–0.690) and hearing function (OR: 0.615; CI: 0.431–0.878). Community and social factors (adjusted for the effects of demographic and physical factors) that were significantly associated with WHO-5 scores were community group (OR: 1.310; CI: 1.003–1.059), community commitment (OR: 1.180; CI: 1.132–1.231), and social networks (OR: 1.525; CI: 1.142–2.037).

## 4. Discussion

In the current study, participants were older adults, certified by the local government as being independent, older people without disabilities, living in Yokohama, the largest city designated by government ordinance in Japan. The demographic characteristics (average age, household composition, and economic status) of our participants were consistent with statistical data reported by the Statistics Bureau of Japan 2022. Therefore, the participants in the current study could be considered representative of Japan’s population of independent older adults.

Firstly, regarding hypothesis 1 in the current study, the results revealed that physical factors of age-related visual and hearing impairments were negatively associated with WHO-5 scores of independent older adults, even after adjusting for the effects of sex and age. The finding of a relationship between WHO-5 scores and visual impairment is consistent with the findings of previous studies [30,31,32,33] reporting that visual impairment in older people has a negative impact on QOL by interfering with activities of daily living. The finding of a relationship between WHO-5 scores and hearing impairment is also consistent with a previous finding [34,35] indicating that hearing impairment in older people has a negative impact on QOL by interfering with interpersonal verbal communication.

These findings indicate the increasing importance of identifying and managing age-related visual and hearing impairments in independent older adults, and the benefits of doing so across SWBs, in addition to mitigating or slowing down the progression of each impairment. Specifically, it is important to support older adults to be aware of their difficulties in vision and hearing and to help them choose the necessary treatment. In addition, emphasis needs to be placed on giving older adults, as well as those around them, proper information about visual and hearing impairment, gradually introducing them to aid use and teaching strategies (e.g., lip reading, nonverbal signals, asking others to speak more clearly) [36].

The results of this study did not show an association between WHO-5 scores and cognitive impairment. Because the subjects of this study were older adults certified by the local government as being independent, healthy, and living in communities, they may have exhibited mild cognitive impairment that did not impair daily life and thus did not reach the diagnostic threshold. There is still a lack of consensus on the relationship between SWB and cognitive function in older adults [37,38,39]. Some studies have reported a persistent association between mental health and perceived cognitive impairment [37], while others have suggested that mental health depends on the social context of stress and support rather than on the degree of cognitive impairment [38]. Further research is needed.

Secondly, regarding hypothesis 2 in the current study, the results revealed that community and social factors were positively associated with WHO-5 scores in independent older adults, even after adjusting for the effects of sex, age, and physical factors. This finding indicates the increasing importance of community and social factors for independent older adults. Strengthening the positive influence of these factors may not only improve the SWB of individual older adults but may also benefit super-aging societies as a whole.

The association between WHO-5 scores and the presence of community groups is consistent with evidence that participation in community groups is associated with well-being after eliminating confounding demographic, socioeconomic, and health-related factors [40,41,42,43]. Community activities in Japan include physical and volunteer activities, hobbies, and activities to improve one’s abilities [44]. It is significant that they were voluntary activities of older adults in Japan and encompass all community-based activities. Groups may offer diverse opportunities for cognitive stimulation, physical activities, social interactions, emotional bonding, collaborative learning, the pursuit of collective goals, and the development of self-esteem, all of which influence mental health [40]. Future research should focus on specific types of community groups that older adults belong to and explore their characteristics relating to SWB [40].

The association between WHO-5 scores and social networks is consistent with evidence that social networks that are supportive have consistently beneficial associations with mental health [45]. Recent studies have reported the potential benefits of social networks and interactions for coping with issues in daily life [46] and fostering emotional well-being [47]. Neighborhood-based social networks may complement the support provided by relatives and friends, which older adults tend to lose as they age [48]. Differing from kinship-based ties founded on marriage or blood, and friendship ties, which need time to build or maintain, neighbors may serve as important public resources [39] that are accessible to every community dweller. In a super-aged society, interventions to maintain or increase neighborhood-based social networks, which are related to SWB, will become increasingly important within public health policies.

The association between WHO-5 scores and community commitment is consistent with evidence of an association between a sense of community and improved health outcomes in older adults, including maintenance of cognitive and mental health [45], halting the progression of frailty [49], and the promotion of healthy aging. As a basic human need, community commitment is not only the basis for solving problems, such as social isolation, loneliness, crime, and accidents within the community, but can also have significant internal effects, which are not visible from the outside, such as the enhanced well-being of older adults [50]. While community commitment is beneficial for enhancing the SWB of older adults, it is also important for building social capital, which is the goal of public health policies in a super-aged society.

The implications for practice of the current study are that it provides empirical data on the factors in the physical functional domain and the community and social domain associated with SWB, which are likely to contribute not only to individual health but also to building a healthy super-aged society. Successful aging [15] involves multidimensional domains. The current findings highlighted the importance of factors in the physical functional domain and the community and social domain in successful aging. Individual factors at the micro level, social circle at the meso level, and larger-scale support at the macro level may be incorporated into a social ecological model. That is, the findings of the current study will allow policy makers, health care professionals, and others to design specific initiatives, tools, and actions to improve SWB among community-dwelling older adults based on the micro level, meso level, and macro level.

The current study had several limitations. First, the cross-sectional design constrained the establishment of a causal relationship between SWB and related factors. Further longitudinal research is needed. Second, the male-to-female ratio: according to official data from the Ministry of Internal Affairs and Communications of Japan, the male-to-female ratio of those aged 75 and over in Japan is about 1:1.8, while the male-to-female ratio in this study is about 1:3. Although we adjusted for gender effects in our analysis, bias may not have been fully adjusted. And lastly, the research field, Yokohama, is the largest ordinance-designated city in Japan, but it is unclear whether the results of this research field can be generalized to suburban and rural areas in Japan or other developed countries.

## 5. Conclusions

This study demonstrated that the following factors were significantly associated with the WHO-5 in independent, healthy, community-dwelling older adults in Japan: visual function, hearing function, community groups, community commitment, and social networks, adjusted for the effects of demographic factors. The current findings indicate that the factors associated with SWB are not only likely to contribute to individual health and longevity, but that they are also important for building a healthy, super-aged society in the future.

## Figures and Tables

**Table 1 healthcare-11-02211-t001:** Demographic characteristics of participants (n = 926).

Variables	Mean	(SD)
Age, years	78.1	(6.7)
The frequency of going out/week	5.2	(1.7)
	n	(%)
Sex		
Male	189	26.0
Female	737	74.0
Living arrangements		
Living alone	264	28.5
Living with spouse	332	35.9
Living with children	136	14.7
Living with spouse and children	122	13.2
Living with children and grandchild	47	5.1
Others	22	2.3
Missing	3	0.3
Economic status		
Sufficient	800	86.4
Insufficient	120	13.0
Missing	6	0.6
Medical treatment history (multiple)		
Presence	757	81.7
Hypertension	450	48.6
Arthralgia	185	20.0
Diabetes	165	17.8
Others	142	15.3
Absence	169	18.3

**Table 2 healthcare-11-02211-t002:** Physical, community and social factors of participants (n = 926).

Variables		
Physical factors	n	%
visual function (impairment)	188	20.4
hearing function (impairment)	157	17.0
cognitive function (impairment)	285	30.8
Community and social factors	n	%
community group (presence)	292	31.6
	mean	(SD)
community commitment (scores)	17.2	(3.7)
social network (scores)	17.5	(6.0)

**Table 3 healthcare-11-02211-t003:** Logistic regression analysis of the related factors in WHO-5 (n = 926).

Variables	SPRC	Odds Ratios (95%CL)
Demographic factors		
age (1: less than mean age 0: more than mean age)	0.708 ***	0.493 (0.352–0.690)
sex (1: female 0: male)	0.180 ***	0.615 (0.431–0.878)
Physical factors ^(a)^		
visual function (1: impairment 0: non-impairment)	0.708 ***	0.493 (0.352–0.690)
hearing function (1: impairment 0: non-impairment)	0.486 ***	0.615 (0.431–0.878)
cognitive function (1: impairment 0: non-impairment)	0.147	1.158 (0.867–1.547)
Community and social factors ^(b)^		
community group (1: presence 0:absence)	1.226 ***	1.310 (1. 003–1.059)
community commitment (1: high 0: low) ^(c)^	1.166 ***	1.180 (1.132–1.231)
social network (1: high 0:low) ^(d)^	1.422 ***	1.525 (1.142–2.037)

Note: ^(a)^ adjusted for demographic factors; ^(b)^ adjusted for demographic factors and physical factors. ^(c)^ cutoff point = mean scores (17.2). ^(d)^ cutoff point = mean scores (17.5). SPRC: standardized partial regression coefficient; social network: community commitment by the Lubben Social Network Scale; community commitment by the Community Commitment Scale. *** *p* < 0.001.

## Data Availability

The data that support the findings of this study are available from Yokohama City Local Government and Yokohama City University but restrictions apply to the availability of these data under the Japan Personal Information Protection Law, which were used under license for the current study, and so are not publicly available. Data are However, available from the first/corresponding authors upon reasonable request and with permission of Yokohama City Local Government and Yokohama City University.

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
