# Peer review of "Subjective Well-Being and Related Factors among Independent, Healthy, Community-Dwelling Older Adults in Japan"

_healthcare, 2023, doi:10.3390/healthcare11152211_

Round 1

Reviewer 1 Report (Previous Reviewer 1)

I have reviewed the paper and I believe that the authors have considered all the comments and made the appropriate changes, which were suggested to them.

I would accept it with those changes for acceptance.

Best regards

Author Response

Thank you for the thoughtful and constructive feedback you provided regarding our manuscript, Subjective well-being and related factors among independent, healthy, community-dwelling older adults in Japan.

Reviewer 2 Report (New Reviewer)

The authors of the paper have identified a gap in the literature concerning the wellbeing of aged people; a pertinent topic given the expected increase in longevity of life.  Using the validated survey from the World Health Organisation (WHO) WHO-5 to survey a preidentified cohort of people in Yokohama, Japan.  The prospective participants were identified by the local government as meeting the requirements of the survey.  The aim of the research was to explore the Subjective Wellbeing (SWB) of independently living Japanese people with their own culture to support research undertaken in other countries and cultures.  All the people identified by the local government were contacted with a request to complete the attached survey regarding their wellbeing and to provide data for the Japanese Government’s “100-year life expectancy strategy” due to the increasing life expectancies of the Japanese and world populations.  The use of a preidentified research cohort enhances the research data as this reduces the bias that can observed in random selection of participants such as wide advertising in age related media.

According to the authors, ‘the purpose of this study was to clarify the associations between SWB and  individual physical factors as well as community and social factors among independent older adults living in the community with an average age of 75 or older, which is expected to increase rapidly in the future identified from an extensive literature search into SWB that had been undertaken by others and published through the WHO organisation so that averages and many of the themes were previously been identified in other cultures’. This research purpose was identified from an extensive literature review which identified previous research reported by the WHO.  Subjective wellbeing is not eudemonic wellbeing and is therefore differentiated.  Two hypotheses were developed around the individual physical factors, and community and social factors.

Following the survey completion, the data was collated and subjected to descriptive statistical analysis to identify dependant variables and proportions of the respondents to each dependant variable.  Complete demographic and correlation statistical analyses were undertaken of the data and compared with the WHO analytical results.  The type of statistical analyses undertaken were appropriate to make meaning of the data which were used to accept or not accept the hypotheses.  The discussion and conclusions are well supported by the data analysis and presented with clarity therefore enhancing reader understanding of the research.

Overall, this is a well thought-out, designed and implemented research project undertaken in collaboration with government agencies, and thus has practical implications for future government planning of communities.  A research project that definitely required publishing.

Author Response

Thank you for the thoughtful and constructive feedback you provided regarding our manuscript, Subjective well-being and related factors among independent, healthy, community-dwelling older adults in Japan.

Reviewer 3 Report (New Reviewer)

To the Author,

Many thanks for providing this manuscript. It was very educational. The purpose of this study is to determine the contribution of physical health issues and social/community engagement to SWB in Japanese seniors. I enjoyed reading the article and respect the research topic chosen. 

The attached file contains comments that can be incorporated to improve the manuscript.

In addition, the following points warrant consideration:

1. Incorporating a sound theoretical model on SWB, physical wellbeing, social wellbeing, and overall health can give the current manuscript a solid theoretical foundation.

2. Individual factors at the micro level, social circle at the meso level, and larger-scale support at the macro level may be incorporated into a social ecological model.

3. The inclusion of clear demographic data in table 1 can enhance reader comprehension.

The discussion is effectively written

Best wishes!

Author Response

Thank you for the thoughtful and constructive feedback you provided regarding our manuscript, Subjective well-being and related factors among independent, healthy, community-dwelling older adults in Japan.   

 ï¼‘.Incorporating a sound theoretical model on SWB, physical wellbeing, social wellbeing, and overall health can give the current manuscript a solid theoretical foundation.

 ⇒In accordance with the reviewer's comments, we have incorporated a successful aging (Rowe,J,W; Kahn,R,L Successful Aging. Gerontologist, 1997) as a sound theoretical model on SWB, physical wellbeing, social wellbeing, and overall health on the revised manuscript: Aging is influenced by the physical, social, and mental well-being, participation in society, protection, security and care. If older adults are able to be physically active and continuing participation in social, economic, cultural, spiritual, and civil affairs, it is ac-tive healthy aging [14]. Moreover, active healthy ageing is the process of optimizing opportunities for successful aging [15]. That is, SWB and its related individual physical factors as well as community and social factors can be the powerful tool for making this successful aging [15]. L.87~

 ï¼’.Individual factors at the micro level, social circle at the meso level, and larger-scale support at the macro level may be incorporated into a social ecological model.

⇒In accordance with the reviewer's comments, We have added to the implications of this study for practice, regarding personal factors at the micro level, social circles at the meso level, and larger-scale support at the macro level.: The implications for practice of the current study that it provides empirical data on the factors in the physical functional domain and the community and social domain associated with SWB, which are likely to contribute not only to individual health but also to building a healthy super-aged society. Successful aging [15] involves multidimensional domains. The current findings highlighted the importance of factors in the physical functional domain and the community and social domain in successful aging. Individual factors at the micro level, social circle at the meso level, and larger-scale support at the macro level may be incorporated into a social ecological model. That is, the findings of the current study will allow policy makers, health care professionals, and others to design specific initiatives, tools, and actions to improve SWB among community-dwelling older adults based on the micro level, meso level, and macro level. L.317~

 ï¼“.The inclusion of clear demographic data in table 1 can enhance reader comprehension.

⇒We have added to Table 1 the details of the medical treatment history (MULTIPLE).

This manuscript is a resubmission of an earlier submission. The following is a list of the peer review reports and author responses from that submission.

Round 1

Reviewer 1 Report

First of all, I would like to congratulate the authors for their study, that with such a large sample, in Japan, they have reached important conclusions that should be taken to a practical treatment, especially in vision and hearing, and their approach to prevent subjective cognitive impairment, and enhance greater personal autonomy.

However, I consider that some small changes should be made in the manuscript, which I indicate below.

Abstract

The authors should not say that the sample is 1900 older adults, since after applying the inclusion criteria and those who actually responded to the survey, there were 926. They should explain it better in the abstract. Or really say that the sample consisted of 926 participants.

Introduction

Lines 64-75 talk about the aging of Japan, this could be moved to the beginning of the introduction, in the second paragraph after lines 35 and 36 where it is said:

 Given the global trend of population aging and in-35 creasing life expectancy, an urgent challenge facing the international community entails.

Methodology

The sample size should be considered with respect to the main outcome variable.

On the other hand, they should expand the information on the tools used, as I indicate line by line.

Line 132

The SDC, specify the acronym of this test and what it measures

Line 134

Mini-Mental State Examination and Frontal Assessment Bat-134 tery scores. A cutoff point of 17/18 was used to determine the presence/absence of mild cognitive impairment in this study; specify what this test measures with respect to the score out of 30 and give more data on sensitivity, specificity, etc.

Lines 141-144

You should specify data on sensitivity and specificity of the scales.

(LSNS-6) and Community Commitment Scale (CCS), or psychometric measures for the population studied.

Discussion

Line 212, since the relationship of cognitive impairment is so strongly linked to the perception of visual and auditory dysfunction, I would like to ask you two questions

1-At the methodological level, it has been proposed that a professional, ophthalmologist and otolaryngologist, determine this visual and auditory acuity, instead of self-reporting (since psychological problems could be influencing?)

2-If you have wanted to highlight how the elderly person lives it, you have considered offering natural therapies, visual and auditory stimulation as a treatment.

I would like you to look for information in this regard and could be added to this section of the discussion.

Line 254, could you justify this section of social and community participation with more bibliographical references, if it depends for example on the type of activities in which the elderly participate, the interests they have and motivate them, or the main occupation to which they have dedicated themselves? All these aspects could generate an added cognitive reserve. They should be contemplated and added in this section.

It is necessary to add the section on limitations and future lines of research where some of the aspects I have mentioned above could be added.

Reviewer 2 Report

Comments

1.     The major problem of the manuscript is that it is not written in an academic manner. In the Introduction part, you need to state the limitations, weakness and gaps of the literature. You mention that there is inconsistency in the relationship between age and subjective well-being. Will your study address the inconsistency? How will you address the inconsistency? Your research purpose (see sentences just before Materials and Methods in page 2) suggest that you look into broader effects of demo characteristics, physiological factors, and community and social factors on subjective well-being. However, you do not allude to it in the Introduction part. I would suggest expanding the section on the research problem. For example, you may point out that

 “There are three limitations and weaknesses in the current literature on the antecedents of subjective well-being. First, ---. Second, ---. Third, ---“.

2.     After you point out these limitations and weakness, put forward your research objectives. “To address these weaknesses and limitations, our study has three objectives. First, ---. Second, ---. Third, ---.”

3.      Have a part of Literature Review and Hypotheses. Build your hypotheses based on literature review. For example (just example!)

Hypothesis 1: Marital status moderates the relationship between age and subjective well-being for retired people. Increasing age is more negatively related to SWB for divorced or widowed/unmarried retired people than for married retired people.

Hypothesis 2: Well cognitively functioned retired people have better subjective well-being than poorly cognitively functioned retired people.

Hypothesis 3: Retired people with wide social network have better subjective well-being than those with narrow social network.

Hypothesis 4: Marital status moderates the relationship between social network and subjective well-being such that the relationship is stronger for divorced or widowed/unmarried than for married retired people.

4.     It is better to measure your variables with 5-or 7-point Likert Scale. You use Yes (1)-or- No (0) style.

5.      For Results, Table 1 is descriptive statistics (mean and SD) and correlations.

6.      Table 2 is the result of regression analysis. You may not need to enter all the variables in the regression.

7.      Discussion.

Since you do not have any hypotheses, the study does not have any focus. You do not attempt to argue for your hypotheses based on any theory and literature review. You do not have any specific results in support of your hypotheses. We do not find any paragraph to discussion any research findings. Thus, I would suggest that you rewrite your Introduction part and have a part of Literature Review and Hypotheses.  
